# An Investigation on Glucuronidation Metabolite Identification, Isozyme Contribution, and Species Differences of GL-V9 In Vitro and In Vivo

**DOI:** 10.3390/molecules24081576

**Published:** 2019-04-22

**Authors:** Han Xing, Dexuan Kong, Chen Ning, Ying Kong, Chang Ren, Yujie Cheng, Hui Cai, Jubo Wang, Di Zhao, Ning Li, Xijing Chen, Zhiyu Li, Yang Lu

**Affiliations:** 1Clinical Pharmacokinetics Laboratory, China Pharmaceutical University, Nanjing 211198, China; xinghankf@126.com (H.X.); kongdexuan22@gmail.com (D.K.); ningchen1996@gmail.com (C.N.); ky0620@126.com (Y.K.); Hazel_Chang@163.com (C.R.); cyj_cpu@163.com (Y.C.); carolhui1001@outlook.com (H.C.); zhaodihehe@126.com (D.Z.); 15895910536@163.com (N.L.); 2Jiangsu Key Laboratory of Drug Design and Optimization, Department of Medicinal Chemistry, China Pharmaceutical University, Nanjing 211198, China; wjb912jh@126.com

**Keywords:** GL-V9, glucuronidation, enzyme kinetics, human recombinant UGTs

## Abstract

GL-V9 is a prominent derivative of wogonin with a wide therapeutic spectrum and potent anti-tumor activity. The metabolism characteristics of GL-V9 remain unclear. This study aimed to clarify the metabolic pathway of GL-V9 and investigate the generation of its glucuronidation metabolites in vitro and in vivo. HPLC-UV-TripleTOF was used to identify metabolites. The main metabolite that we found was chemically synthesized and the synthetic metabolite was utilized as standard substance for the subsequent metabolism studies of GL-V9, including enzyme kinetics in liver microsomes of five different species and reaction phenotyping metabolism using 12 recombinant human UDP-glucuronosyltransferase (UGT) isoforms. Results indicated that the glucuronidation reaction occurred at C5-OH group, and 5-*O*-glucuronide GL-V9 is the only glucuronide metabolite and major phase II metabolite of GL-V9. Among 12 recombinant human UGTs, rUGT1A9 showed the strongest catalytic capacity for the glucuronidation reaction of GL-V9. rUGT1A7 and rUGT1A8 were also involved in the glucuronidation metabolism. K_m_ of rUGT1A7-1A9 was 3.25 ± 0.29, 13.92 ± 1.05, and 4.72 ± 0.28 μM, respectively. In conclusion, 5-*O*-glucuronide GL-V9 is the dominant phase II metabolite of GL-V9 in vivo and in vitro, whose formation rate and efficiency are closely related to isoform-specific metabolism profiles and the distribution of UGTs in different tissues of different species.

## 1. Introduction

Flavonoids as a large subgroup of phenolic metabolites that are widely present in plants used as folk medicines [1]. Also, they are found in tea, wine [2], propolis and honey [3], and thus commonly account for a major component of human diet [4]. Flavonoids and their derivatives have been considered as potential therapeutic compounds for a multitude of diseases because of their wide range of biological and pharmacological activities, including oestrogenic activity [5], anti-inflammatory activity [6], antioxidant activity [7], antiallergic activity [8,9], anti-tumor activity [10,11] and vascular regulation activity [12,13]. As a versatile source of antitumor drugs, flavonoids have consistently attracted increasing attention of researchers. However, the pharmacokinetic properties of many flavonoids, such as wogonin, are rather unsatisfactory [14]. This explains the fact that only a few flavonoids are clinically used currently. Therefore, a series of derivatives have been designed and synthesized to improve the pharmacokinetic profiles and promote the bench-to-bedside translation of flavonoids [15,16,17,18]. 

GL-V9 is a wogonin derivative with strong tumor growth inhibitory effects in vitro and in vivo. In human hepatocellular carcinoma, GL-V9 could induce G2/M cell cycle arrest and mitochondrial-mediated apoptosis [15]. In gastric cancer, GL-V9 exhibited significant anti-tumor effects in xenograft model through a calcium-mediated mitochondrial apoptotic pathway with satisfactory safety profile [19]. In breast carcinoma, the inhibitory effects of GL-V9 was verified by down-regulating the expression and activity of matrix metalloproteinase-2/9 [20]. Although GL-V9 acts similarly to wogonin in terms of tumor inhibitory mechanism, the prominent advantage of GL-V9 over wogonin is that it achieves the same anti-tumor effect at much lower concentrations, suggesting that GL-V9 is a more promising candidate for cancer treatment.

The bioavailability of most flavonoids is undesirable to be developed as oral drugs due to their extensive metabolism or rapid excretion [21,22,23,24,25,26]. Glucuronidation is the major metabolic pathway of a variety of exogenous compounds, including flavonoids, and it probably affects the bioavailability, and even biological activities or toxicities [26]. Although most of flavonoids are metabolized quickly after administration and thus exhibit poor bioavailability, some glucuronidation metabolites of flavonoids also possess therapeutic bioactivities [27] and have the potential to be developed into drugs. Thus, metabolite identification is an important part in the development of these polyphenolic compounds. As GL-V9 is the derivative of wogonin, whose glucuronidation metabolite (wogonoside) showed significant anti-tumor activity in various cancer models [28,29,30], it is conjectured that GL-V9 might share the similar metabolic process with wogonin and its glucuronide conjugates might possess therapeutic activities as wogonoside. Therefore, systemic investigation on glucuronidation profiles of GL-V9 will be beneficial regarding a comprehensive understanding on biological effects of GL-V9 and its metabolites.

UDP-Glucuronosyltransferase (UGT) superfamily is a kind of major phase II metabolism enzymes expressed in human [31]. More than 35% of all drugs are metabolized via UGT-mediated glucuronidation pathway [32]. The contribution of the different UGT isoforms to drug metabolism is highly variable due to different substrate specificity and different expression levels in organs. Generally, glucuronidation studies of a compound are divided into several aspects [33], including UGT-isoform-specific metabolic fingerprint (GSMF) [34], enzyme kinetics and the evaluation of the importance of glucuronidation in the metabolism of the compounds. Previous studies showed that GL-V9 underwent an extensive metabolism in liver after administration, where it could be catalyzed by a family of UGT [35]. It has been reported that UGT1A1, UGT1A3, UGT1A4 and UGT1A9 genes are widely expressed in human liver [36]. Among the UGT1A subfamily, UGT1A1 (51.60%) and UGT 1A9 (43.67%) are the two most abundant isoforms in human liver [37]. By means of commercial recombinant enzymes and microsomes from different species, the kinetics parameters of GL-V9 can be obtained to evaluate its metabolic pathway.

The objective of the current study is to illustrate the glucuronidation characteristics of GL-V9 in vivo and in vitro. Combined with its special structural characteristics, glucuronidation metabolite of GL-V9 was identified as 5-*O*-glucuronide GL-V9 using HPLC-UV-TripleTOF. Next, the chemically synthesized 5-*O*-glucuronide GL-V9 was prepared and its structure was confirmed using ^1^H-nuclear magnetic resonance (^1^H-NMR). This synthetic metabolite was used as standard substance in the following experiments. Enzyme kinetics of formation of 5-*O*-glucuronide GL-V9 in five different species was studied using different liver microsomes. The metabolism profiles of GL-V9 in human liver microsomes (HLMs) play an important role in predicting the disposition for clinical application. Twelve recombination UGT isoforms were applied to investigate the contribution of isozymes to UGT-catalyzed phase II metabolism of GL-V9. The pharmacokinetic study of GL-V9 and its glucuronide metabolite in rats were also performed to evaluate the role of glucuronidation in the metabolism of GL-V9 in vivo.

## 2. Results

### 2.1. Identification of Glucuronidation of GL-V9 In Vitro and In Vivo

HPLC-UV-TripleTOF analysis indicated that glucuronidation conjugates were formed in the presence of liver microsomes and UDPGA in vitro. The result of formation of 5-*O*-glucuronide GL-V9 in HLMs incubation system was shown in Figure 1A,B. According to the UV detection results, the retention times of GL-V9 and 5-*O*-glucuronide GL-V9 were 9.31 min and 10.50 min, respectively. The next step was to further characterize the chromatographic and mass spectral behavior of GL-V9. Thus, the structure of 5-*O*-glucuronide GL-V9 can be predicted on the basis of the analytical data, and the molecular formula was confirmed as C_30_H_36_O_11_N by the TripleTOF MS spectrum (*m*/*z* = 586.2279 (−0.7 ppm) [M + H]^+^). The spectrum was displayed in Figure 1C and the fragment ions of 5-*O*-glucuronide GL-V9 were *m*/*z* 410.2 and *m*/*z* 126.1 produced by deglycosylation and the cracking of the 7-heterocyclic side chains in positive ion mode.

The metabolite identification in vivo showed similar results. Compared with the results of in vitro analysis, the glucuronidation metabolites of GL-V9 in rat plasma presented the same retention time and fragment ions (Figure 2). There was an apparent decrease of peak area in the samples containing β-glucuronidase, thus it was confirmed that this phase II metabolite of GL-V9 was produced by a glucuronidation reaction.

### 2.2. Chemical Synthesis of 5-O-Glucuronide GL-V9 and ^1^H-NMR Analysis

To further investigate the metabolic pathway of GL-V9, 5-*O*-glucuronide GL-V9 was chemically synthesized. The structure was confirmed using ^1^H-NMR as described in the Materials and Methods, and the purity of the conjugate was analyzed using HPLC-UV to be > 95%.

The retention time of synthesized 5-*O*-glucuronide GL-V9 was consistent with the analyte observed in vitro and in vivo (Figure 1 and Figure 2). In order to further clarify the hypothesis that 5-*O*-glucuronide GL-V9 was the glucuronide metabolite of GL-V9, a rat plasma sample spiked with authentic standard of 5-*O*-glucuronide GL-V9 was analyzed using HPLC-UV-TripleTOF. As shown in Figure 3, the peak area increased obviously compared with the control sample without synthesized metabolite from 2.3215 × 10^5^ to 6.5219 × 10^5^ (Figure 3A,B). These experiments proved that 5-*O*-glucuronide GL-V9 was the correct synthetic product, consistent with the results in vivo and in vitro, and can be used as the standard substance of glucuronide metabolite in the following experiments.

### 2.3. Main UGT Isoforms Responsible for the Metabolism of GL-V9 In Vitro

In order to identify the main isoform(s) of UGTs involved in the glucuronidation metabolism of GL-V9, twelve recombinant human UGT isoforms (rUGT1A1, rUGT1A3, rUGT1A4, rUGT1A6, rUGT1A7, rUGT1A8, rUGT1A9, rUGT1A10, rUGT2B4, rUGT2B7, rUGT2B15 and rUGT2B17) were employed. GL-V9 at three concentrations (0.5, 1, 5 μM) were incubated with recombinant UGTs individually at 37 °C for 1 h. Incubation time was selected based on determination of reaction velocity linearity. Glucuronidation rates of different recombinant UGTs were described in Figure 4 and there were significant differences among these 12 isozymes (*p* < 0.05). rUGT1A9 played a key role in glucuronidation of GL-V9 with the most rapid rates of formation (6.81 ± 0.46 to 57.13 ± 1.20 pmol/min/mg). Moreover, rUGT1A7 and rUGT1A8 were also involved in the conjugation reactions, and the glucuronidation rates were 1.90 ± 0.07 to 8.08 ± 0.56 pmol/min/mg and 0.72 ± 0.01 to 6.16 ± 0.20 pmol/min/mg respectively. rUGT1A1 showed minor catalytic ability with glucuronidation rate ranged from 0.20 ± 0.01 to 1.66 ± 0.07 pmol/min/mg. Among three chosen concentrations of GL-V9 in incubation systems, rUGT1A9 was always the top one in the catalytic capacity of the glucuronidation reaction. As for other rUGT1As (1A3, 1A4, 1A6, 1A10), the metabolism rates were too low to be biologically relevant. Similarly, rUGT2Bs (2B4, 2B7, 2B15, 2B17) did not show apparent catalytic activity in the production of 5-*O*-glucuronide GL-V9.

### 2.4. Kinetics of GL-V9 in Liver Microsomes and Recombinant UGTs (1A7, 1A8 and 1A9)

Kinetics studies of GL-V9 were conducted over a range of substrate concentrations in vitro using monkey, dog, rat, mice and human liver microsomes (MLMs, DLMs, RLMs, MiceLMs and HLMs) and rUGTs (rUGT1A7, 1A8 and 1A9). As demonstrated in Figure 5, liver microsomes-mediated glucuronidation of all five species exhibited autoactivation kinetics. Enzyme kinetic parameters of the formation of 5-*O*-glucuronide GL-V9 were summarized in Table 1. The K_m_ values in monkey, dog, and human liver microsomes showed no significant difference (11.19 ± 0.47, 12.48 ± 1.43, 11.15 ± 0.66 μM), whereas a much higher K_m_ value was observed as 20.12 ± 1.04 μM in MiceLMs. Among all these liver microsomes, MLMs exhibited the highest maximum velocity (V_max_) than others at 182.5 ± 4.92 pmol/min/mg, while the V_max_ values of DLMs and RLMs were only 22.03 ± 0.42 and 32.72 ± 1.45 pmol/min/mg, respectively. And the maximum velocity value in HLMs (88.49 ± 4.21 pmol/min/mg) was only half of that in MLMs. In the case of intrinsic clearance (Cl_int_, V_max_/K_m_) of 5-*O*-glucuronide GL-V9, there were three levels among these different species liver microsomes. The highest Cl_int_ value was presented in MLMs (16.31 μL/min/mg), which was approximately 9-fold higher than that in DLMs (1.76 μL/min/mg). Nevertheless, the Cl_int_ of 5-*O*-glucuronide GL-V9 in MiceLMs, RLMs and HLMs were at a moderate level and relatively similar (7.61 μL/min/mg for MiceLMs, 5.00 μL/min/mg for RLMs, 7.94 μL/min/mg for HLMs). 

rUGTs-mediated glucuronidation also exhibited autoactivation kinetics as shown in Figure 6. Table 2 summarizes the enzyme kinetics parameters of rUGT1A7, 1A8 and 1A9. The K_m_ values of 5-*O*-glucuronide GL-V9 of rUGT1A7 and 1A9 were similar (3.25 ± 0.29, and 4.72 ± 0.28 μM, respectively), while rUGT1A8 showed the highest K_m_ values as 13.92 ± 1.05 μM. Among these three isozymes, the highest V_max_ was observed in rUGT1A9 (148.1 ± 2.56 pmol/min/mg), while the V_max_ values of rUGT1A7 and rUGT1A8 were 31.15 ± 0.65 and 50.63 ± 4.09 pmol/min/mg, respectively. In the case of intrinsic clearance (Cl_int_, V_max_/K_m_), there were significant differences among rUGT1A7, 1A8, 1A9 (*p* < 0.05) The highest Cl_int_ was observed in rUGT1A9 (31.38 μL/min/mg), followed by rUGT1A7 (9.58 μL/min/mg), and rUGT1A8 showed the lowest Cl_int_ value (3.64 μL/min/mg) which was nearly one ninth of that in rUGT1A9. 

### 2.5. Quantification of GL-V9 and 5-O-glucuronide GL-V9 In Vivo

The quantification of GL-V9 and 5-*O*-glucuronide GL-V9 in vivo was carried out using UPLC-MS/MS. All the plasma samples were collected from rats receiving GL-V9 by the means of oral delivery (gavage, 50 mg/kg). The retention time of GL-V9 and 5-*O*-glucuronide GL-V9 was 2.62 and 2.84 min, respectively. The mean plasma concentration of GL-V9 and 5-*O*-glucuronide GL-V9 versus time profiles were displayed in Figure 7. The pharmacokinetic parameters were processed by WinNonlin (Version 7.1, Pharsight, Mountain View, CA, USA) using non-compartmental model. The t_1/2_ and C_max_ of GL-V9 was 3.08 ± 1.05 h and 167.33 ± 50.05 ng/mL (Appendix A) [38], while that in 5-*O*-glucuronide GL-V9 was 5.71 ± 2.53 h and 325.67 ± 54.53 ng/mL. After oral administration, the AUC_0-t_ of GL-V9 and 5-*O*-glucuronide GL-V9 was 1212.06 ± 246.47 and 4220.70 ± 577.22 h*ng/mL, respectively. The metabolism fraction (f_m_) of 5-*O*-glucuronide GL-V9 was calculated as 87.39 ± 12.37% using Equation (1): (1)$$\frac{AUC_{metabolite}}{AUC}=f_{m}\frac{CL}{CL_{metabolite}}$$

## 3. Discussion

Flavonoids are extensively distributed in plants and foods in Nature, sharing the same structural backbone. Glucuronidation is a vital pathway for metabolism and clearance of flavonoids and their derivatives containing phenolic hydroxyl groups and/or methoxy groups [39]. However, the regioselectivity of glucuronidation of flavonoids largely depends on the location of substitutional groups [40]. Furthermore, the biological activity may be influenced by the complicated structure modification. The relationship between the specific position and the possibility of glucuronidation has been systematically studied and the effect of individual UGT isoforms on that has also been taken into account and investigated. Interestingly, studies on the production of glucuronidation metabolites of quercetin, a typical flavonoid, indicated that glucuronidation reaction did not intend to occur at 5-position in flavonoid ring [41], which suggested that 5-position might be a null-glucuronidation site. In fact, the existence of 4-carbonyl group and bulky structure of glucuronic acid would hinder the conjunction reaction at this site, making 5-OH group less accessible for glucuronidation in the presence of other available hydroxyl groups. However, the further exploration of possible glucuronidation position demonstrated that 5-*O*-glucuronidation of monohydroxy-flavonoes can be performed although their reactivity was the lowest [42]. Li et al. also reported the presence of a 5-*O*-glucuronide metabolite of wogonin in SD rats [43], confirming the reactivity of the 5-OH in flavonoids. As for the GL-V9, it shares the same backbone with wogonin while it possesses only one 5-OH group in the structure, thus the glucuronide conjunction, if it happens, can only occur at this position. The result of the present study is in accordance with this hypothesis, and the 5-*O*-glucuronide GL-V9 was formed both in vivo and in vitro as the glucuronic acid metabolite of GL-V9. Besides, no other phase II conjugate, e.g., sulfates and methylated metabolites, were detected from in vivo nor in vitro samples, suggesting the unlikelihood of these metabolic pathways. The metabolism fraction of 5-*O*-glucuronide GL-V9 in rat was 87.39 ± 12.37%, calculated using the in vivo data from metabolite quantification study in rats after oral administration. Accordingly, we demonstrated that 5-*O*-glucuronide GL-V9 was the main phase II metabolite of GL-V9 in vivo. 

Next, to further investigate the metabolic profiles of GL-V9, the process of glucuronidation was explored in liver microsomes derived from five species (monkey, dog, rat, mouse, and human). As the results show, significant difference among these species in glucuronidation of GL-V9 was observed. The glucuronidation rate in MLMs was the fastest which was >8-fold faster than that in DLMs. Among these tested animal species, C_lint_ value in MiceLMs was closest with that in HLMs, indicating that the mouse is an appropriate human relevant model to study glucuronidation of GL-V9 in vivo. Comparisons on the glucuronidation of GL-V9 help us to better understand the characters of species-dependent metabolism. These observed differences might closely tie to the expression and the capacity of UGTs in microsomes from different species. 

Previous studies have shown that each flavone has a distinctive UGT-isoform-specific metabolic fingerprint (GSMF) closely related to the structure [34]. The current task was to define the metabolic characteristics of GL-V9 mediated by the application of commercial human recombinant UGT isoforms. In the glucuronidation studies with 12 recombinant UGT isoforms, rUGT1A9 showed the top catalytic activity with the most rapid formation rate followed by rUGT1A7 and rUGT1A8, while other isozymes did not exhibit evident catalytic activity. These results clearly indicated that rUGT1A9 played a key role, and along with rUGT1A7 and rUGT1A8 mainly shared the responsibilities of glucuronidation metabolism of GL-V9. Actually, in human liver, UGT1A1, UGT1A4 and UGT1A9 are reported to be expressed highly at mRNA level [37]. In mouse liver, the expression of Ugt1a5 and Ugt1a6 is also higher than other isoforms besides Ugt1a1 and Ugt1a9 [44]. Additional reports indicated that the UGTs genes among different species possess homology which leads to a relatively high similarity of UGTs in humans and rodents [44]. Therefore, the diversity of UGTs’ expression in five species may be the main reason for the differences in glucuronidation rates. It is worth noting that catalytic efficiency and the amount of expression of independent UGTs in different species should also be considered even in the case that the isoforms may be the same. On the other hand, studies about DSMF help us to further predict the safety and effectiveness of GL-V9, considering the potential drug-drug interactions [34].

To determine the glucuronidation metabolism of GL-V9 in vivo, plasma samples were obtained from rats after oral administration (gavage, 50 mg/kg). As shown in Figure 7, formation of 5-*O*-glucuronide GL-V9 generally corresponded with the elimination of parent GL-V9 at each measured time point. These results indicated that GL-V9 underwent a rapid and extensive biotransformation to 5-*O*-glucuronide upon absorption. The production of metabolite depended on the increase of GL-V9 within the first 20 min post-dose. In the next 12 h, the metabolic capacity of liver dominated the generation of 5-*O*-glucuronide GL-V9. Results of the present study showed that glucuronidation was an important and main metabolic pathway for GL-V9 in vivo and in vitro.

## 4. Materials and Methods

### 4.1. Chemical and Reagents

GL-V9 (C_24_H_27_NO_5_, MW: 409.47, purity > 99.3%) was synthesized by Prof. Zhiyu Li (China Pharmaceutical University, Nanjing, China). Caffeine (used as internal standard, purity ≥ 99%) was purchased from Sigma (Munich, Germany). Methanol and acetonitrile were of HPLC grade commercially available (Tedia, Fairfield, OH, USA). Formic acid (FA) and ammonium acetate were acquired from Lingfeng Company (Shanghai, China) and Xilong Scientific Company (Shantou, China), respectively. Dimethyl sulfoxide (DMSO) used in this study was from Sinopharm Chemical Reagent Co., Ltd. (Shanghai, China). Hydroxypropyl-β-cyclodextrin (HP-β-CD) was provided by J&K^®^ (Shanghai, China). The pooled RLMs (rat liver microsomes), MiceLMs (mouse liver microsomes), DLMs (beagle dog liver microsomes), HLMs (human liver microsomes), and MLMs (monkey liver microsomes) were obtained from RLD (Research Institute for Liver Disease Co., Ltd., Shanghai, China). Human recombinant UGT isoforms (rUGT1A1, rUGT1A3, rUGT1A4, rUGT1A6, rUGT1A7, rUGT1A8, rUGT1A9, rUGT1A10, rUGT2B4, rUGT2B7, rUGT2B15 and rUGT2B17) were purchased from Corning (Woburn, MA, USA). Uridine 5′-diphosphoglucuronic acid (UDPGA, trisodium salt) was purchased from Nacalai Tesque, Inc (Kyoto, Japan). Hydrochloric acid and lactic acid were of analytical grade purchased from Nanjing Chemical Reagent Co., Ltd. (Nanjing, China). Tris base was obtained from Nanjing SunShine Biotechnology Co., Ltd. (Nanjing, China). Purified water used for UPLC-MS/MS and HPLC-UV-TripleTOF analysis was produced by a Milli-Q water purification system (Millipore, Bedford, MA, USA).

### 4.2. Animals

Male Sprague-Dawley rats (280–350 g, provided by the Qinglong Mountain animal breeding ground, Nanjing, China) were housed in cages under controlled conditions (12 h light/dark cycle, temperature maintained at 24 ± 1 °C and humidity kept at 50%) for several days. Animals were raised with free access to food and water before experiment. All animal procedures were in accordance with the rules of Animal Ethics Committee of China Pharmaceutical University (No. SYXK-(Su)-2016-0011).

### 4.3. Identification of Glucuronidation of GL-V9 In Vivo and In Vitro

Blood samples of rats were obtained from orbital venous plexus after oral administration of GL-V9 (gavage, 100 mg/kg). Methanol was added to precipitate protein and to extract GL-V9 and its metabolites. After vortex and centrifugation at 21,500× *g* for 10 min, 200 μL supernatant was transferred to a tube and evaporated to dryness. Then 50 μL pure methanol was added again to dissolve and enrich the analytes before HPLC-UV-TripleTOF analysis.

The composition of the incubation system (200 μL) contained liver microsomes of different species (0.1 mg protein/mL), 5 mM saccharolactone (D-saccharic acid 1,4-lactone monohydrate), 10 mM MgCl_2_, alamethicin (50 μg/mg protein), Tris-HCl buffer (50 mM, pH 7.4) and GL-V9 (1 μM). After 5 min of preincubation at 37 °C, UDPGA (5 mM) was added to initiate the phase II reaction. Control incubation samples were performed without UDPGA or without substrates (GL-V9). The reaction was terminated by addition of ice-cold methanol (three volumes) after incubation for 1 h at 37 °C. Samples were vortexed, centrifuged at 21,500× *g* for 10 min and then stored at 4 °C until HPLC-UV-TripleTOF analysis. To confirm that these metabolites are conjugates resulting from glucuronidation reaction, enzymatic hydrolysis of samples was processed. β-glucuronidase (100 KU, Sigma) was added and the hydrolysis was performed overnight at 37 °C. The control samples were carried out without β-glucuronidase under the same condition.

### 4.4. Chemical Synthesis of 5-O-glucuronide GL-V9 and NMR Analysis

The synthetic route is shown in Scheme 1. GL-V9 (1 g, 2.4 mmol), compound **1** (0.97 g, 2.4 mmol), silver oxide (0.56 g, 2.4 mmol), and quinoline (10 mL) were added to a 10 mL round-bottom flask under the protection of nitrogen. The reaction mixture was stirred at room temperature for 24 h. After stopped, it was filtered. The filtrate was concentrated under reduced pressure to remove quinoline. Then 5 mL of 1 M NaOH aqueous solution was added to the residual, and the mixture was stirred at 10 °C for another 2 h. At the end of reaction, KH_2_PO_4_ was added to adjust the pH to 5~6. The mixture was then subjected to preparative-HPLC. The peak of the target product was collected, and concentrated under reduced pressure to afford the product as yellow solid (5 mg, Yield = 0.3%). ^1^H-NMR (300 MHz, CD_3_OD-_D2O_): *δ* = 7.82 (d, 2H, Ar-H), 7.43 (d, 2H, Ar-H), 7.01 (s, 1H, Ar-H), 6.59 (s, 1H, Ar-H), 4.80 (d, 1H, OCH-O), 4.14 (m, 2H, OCH2), 3.82 (s, 3H, OCH3), 3.50~3.82 (m, 4H, CH-O), 2.96~3.22 (m, 6H, CH2-N), 1.79~2.08 (m, 8H, CH2-CH2) ppm.

To further confirm that the glucuronidation GL-V9 is the metabolites in rat plasma, chemically synthesized GL-V9 was added to rat plasma samples. According to the analysis of HPLC-UV-TripleTOF, the retention time, peak area and cracking ion pairs of analytes were compared.

### 4.5. Glucuronidation of GL-V9 in Monkey, Beagle Dog, Rat, Mouse and Human Liver Microsomes

Enzyme kinetics experiments were carried out in triplicate among five different species, including monkey, beagle dog, rat, mouse, and human. The reaction system and operations are as same as described in Section 4.3. The solutions of GL-V9 were selected over a range of concentrations (0.05, 0.2, 0.5, 1, 2, 5, 10, 20 μM). According to the different metabolic capacity, the protein concentration of liver microsomes of different species is different as shown in Table 1.

### 4.6. Quantification of GL-V9 and Its Metabolites in Rat Plasma

To study the pharmacokinetic characteristics of GL-V9 and its metabolites in rats, GL-V9 was administered by oral delivery (gavage, 50 mg/kg). Blood samples were obtained from orbital venous plexus at different time points (0.083, 0.17, 0.33, 0.75, 1, 1.5, 2, 4, 6, 8, 10, 12, 16, 24 h). 50 μL sample was prepared by adding 200 μL ice-cold methanol containing 500 ng/mL internal standard (caffeine), and supernatants were withdrawn for quantification of GL-V9 and 5-*O*-glucuronide GL-V9 using UPLC-MS/MS method, simultaneously. Control group plasma samples were obtained from rats with saline administration.

### 4.7. UGT Phenotyping and Enzymes Kinetic Studies of rUGTs (rUGT1A1, 1A7, 1A8, 1A9)

Twelve human recombinant UGT isoforms (rUGT1A1, rUGT1A3, rUGT1A4, rUGT1A6, rUGT1A7, rUGT1A8, rUGT1A9, rUGT1A10, rUGT2B4, rUGT2B7, rUGT2B15, and rUGT2B17) were employed to identify the main UGT enzyme(s) responsible for glucuronidation of GL-V9 under mixture conditions as described in Section 4.3. Three substrate concentrations (0.5, 1, 5 μM) were chosen and prepared in the system and two final protein concentration (0.2 and 0.25 mg/mL) were performed by catalytic activities of individual rUGTs. After being terminated by addition of 150 μL methanol containing IS (caffeine, 500 ng/mL), samples were performed and stored at −70 °C before analysis.

For recombinant enzyme kinetic experiments, serial concentrations of GL-V9 (0.05, 0.2, 0.5, 1, 2, 5, 10, 20 μM) were prepared and incubated with rUGT1A7, 1A8 or 1A9 for 1 h at 37 °C, which showed the main effect in glucuronidation of GL-V9 in the former experiment.

Rates of the glucuronidation of GL-V9 were described as amounts of glucuronides formed per minute per milligram (pmol/min/mg). The calculation model of kinetic parameters was selected by the profile of Eadie-Hofstee plots. It often includes three possibilities: (1) the Eadie-Hofstee plot was linear, the formation rates of glucuronides (V) at different substrate concentrations (C) were fit to the standard Michaelis-Menten calculation of GraphPad Prism (version 6.02, GraphPad Software Inc., Guildford, Surrey, UK); (2) the Eadie-Hofstee plot was atypical kinetic-autoactivation, the rates and concentration were fit with the allosteric sigmoidal function of GraphPad Prism; (3) the Eadie-Hofstee plot was atypical kinetic-biphasic, the rates and concentration were fit with the k_cat_ option of GraphPad Prism. 

### 4.8. Analytical Conditions

#### 4.8.1. HPLC-UV-TripleTOF Methods

Glucuronide conjugates identification was conducted using HPLC-UV-TripleTOF system. Chromatography: 20 μL samples were loaded into the HPLC system (LC-20A, Shimadzu, Kyoto, Japan)–UV (SPD-10A, ShimadzU) and separated by a C_18_ column (150 × 2.0 mm, 2.1μm, Shimadzu, VP-ODS) at a flow rate of 0.3 mL/min for 30 min. A: Water (0.1% formic acid) and B: acetonitrile was more suitable as elution mobile phase for GL-V9 and metabolites. The gradient was as described below: 0.01 min (5% B)–2.0 min (5% B), 17.0 min (70% B)–20.0 min (95% B), 25.0 min (95% B)–27.0 min (5% B), 30.0 min (stop). The wavelength of UV detection was set at 370 nm.

Mass Spectrometry: The eluent analytes were detected using TripleTOF5600 TM system under the proper conditions. Samples were analyzed using a curtain gas of 25 psi, ion source gas 1 and 2 of 20 psi and 15 psi, respectively. The ionspray voltage floating (ISVF) was set of 5500 V, and the heater interface temperature was 500 °C. The declustering potential (DP) and collision energy (CE) was 100.0 V and 28.0 eV, respectively. The TripleTOF 5600TM system operated in the information dependent acquisition (IDA) mode which was made use of a TOF/MS scan type with an accumulation time of 0.25 s. The TOF masses ranged from 100 to 1000 Da.

#### 4.8.2. UPLC-MS/MS Methods

An ultra-high performance liquid chromatography (UPLC) system (LC-30A, Shimadzu) and a triple quadruple tandem mass spectrometer (AB SCIEX 4000, AB SCIEX, Foster City, CA, USA) were applied to determine GL-V9 and the glucuronide of GL-V9, simultaneously. A C_18_ column (150 × 2.0 mm, 2.1 μm, Shimadzu, VP-ODS) was used to separate the GL-V9 and 5-*O*-glucuronide GL-V9 in biological samples. The mobile phase comprised of acetonitrile (A) and water (B: 5 mM ammonium acetate and 0.1% formic acid). The gradient elution method was designed as follows: 0.01 min (10% A)–0.2 min (10% A), 0.5 min (40% A)–2.2 min (55% A), 2.5 min (70% A)–3.0 min (70% A), 3.5 min (10% A)–4.5 min (stop) at flow rate of 0.30 mL/min. Analytes were detected by electrospray positive ionization and multiple reactions monitoring (MRM). The ion transitions of GL-V9, 5-*O*-glucuronide GL-V9 and IS (caffeine) were *m*/*z* 410.20→126.10, 586.00→410.00, and 195.1→138.0 *m*/*z*, respectively, and their retention time was 2.79, 2.38, and 2.37 min.

### 4.9. Statistical Analysis

All the data in present study are expressed as the mean ± SD of usually three independent experiments. Variables were regarded as normally distributed and statistical analyses were performed by one-way ANOVA with Tukey test for multiple comparisons using SPSS Statistics 19 (SPSS Inc., Chicago, IL, USA). The values indicated to be significantly different when *p* < 0.05.

## 5. Conclusions

In summary, a novel GL-V9 phase II metabolite was identified and synthesized for the first time. Detailed metabolism studies demonstrated that glucuronidation at 5-position is an important metabolic pathway for GL-V9, and 5-*O*-glucuronide GL-V9 is the major phase II metabolite of GL-V9 in vivo and in vitro. UGT isoform-specific metabolism is the best way to investigate the enzyme kinetic profiles of GL-V9 glucuronidation. From quantitative point of view, rUGT1A7, 1A8, and 1A9 possess higher glucuronidation rate and efficiency than others in the metabolism of GL-V9.

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
