# Peer review of "An Investigation on Glucuronidation Metabolite Identification, Isozyme Contribution, and Species Differences of GL-V9 In Vitro and In Vivo"

_molecules, 2019, doi:10.3390/molecules24081576_

Round 1

Reviewer 1 Report

Summary:

The authors show more details about the metabolism of GLV9 through modifications of the enzyme that catalyzes its transformation. But they do not expose with sufficient clarity what consequences, their study has besides that the enzymatic process can occur with isoforms of the enzyme UGT

I have some minor comments that would be useful to improve the clarity of the work

Line 13-14: One of the “anticancer” can be remove

Line 20: The abbreviation UGTs, can be explain to the text

Line 331-335: the author’s don´t wrote nothing about the temperature of reaction and about the volume

Author Response

Response to the reviewer 1 comments:

The authors show more details about the metabolism of GLV9 through modifications of the enzyme that catalyzes its transformation. But they do not expose with sufficient clarity what consequences, their study has besides that the enzymatic process can occur with isoforms of the enzyme UGT

I have some minor comments that would be useful to improve the clarity of the work.

Point 1: Line 13-14: One of the “anticancer” can be remove

Response 1: We are sorry for the incorrect writing. We have revised the manuscript according to the reviewer’s comments.

Point 2: Line 20: The abbreviation UGTs, can be explain to the text

Response 2: We highly appreciate the valuable comment from the reviewer. The explanation for “UGTs” has been added at the site it first occurred in the manuscript.

Point 3: Line 331-335: the author’s don´t wrote nothing about the temperature of reaction and about the volume

Response 3: We highly appreciate the constructive suggestion and feel sorry for our negligence. The description about the incubation system including temperature (37 ) and volume (200 μL) has been added in the “4.3. Identification of Glucuronidation of GL-V9 in vivo and in vitro”. All the changes are highlighted in the revised manuscript.

Reviewer 2 Report

This manuscript needs to be edited in English. Too many inaccurate usage of words and phrases. What is a prominent derivative? metabolism properties should be metabolism pathways. The title is too long and confusing, not only in English, but also in content. I only see in vitro results reported here, why it is written as in vitro and in vivo everywhere? 

Author Response

Response to the reviewer 2 comments:

Point 1: This manuscript needs to be edited in English. Too many inaccurate usage of words and phrases. What is a prominent derivative? metabolism properties should be metabolism pathways. The title is too long and confusing, not only in English, but also in content. I only see in vitro results reported here, why it is written as in vitro and in vivo everywhere?

Response 1: We highly appreciate the reviewer’s constructive suggestions, which aims at improving the quality of this manuscript. We have carefully revised the manuscript and corrected the inaccurate usage according to the reviewer’s suggestions about English writing. We also changed the title as “Investigation on Glucuronidation Metabolite Identification, Isozyme Contribution, and Species Differences of GL-V9 In Vitro and In Vivo”, hoping it’s easy-understanding for readers.

The aim of this study is to investigate the glucuronidation metabolism of GL-V9 both in vitro and in vivo, including metabolite identification and synthesis, metabolite quantification, and kinetic profiles of different liver microsomes and UGT isozymes. As for the in vivo studies, we reported the results about in vivo metabolite identification and quantification in rat after oral administration of GL-V9 in the part “2.1” and “2.5”.

Reviewer 3 Report

The current manuscript describes the identification and quantification of a glucuronide metabolite of GL-V9 in vitro and in vivo and investigation of the major isoforms of 5 UDP-glucuronosyltransferase (UGT) involved. The manuscript seems to be well written and methodologically sound. However, some points should be clarified or ameliorated:

Section 2.5: other pharmacokinetic parameters should be presented (e.g . Cmax, tmax, AUC0-inf, t1/el, etc). How was performed the oral delivery? Which was the reason to select the oral delivery as it was expected a low bioavailability?

Section 4.3: the centrifugation speed should be in g (not in rpm).

Section 4.8: which was the statistical test used, one-way ANOVA or student-t test? The authors tested for normal distribution? When variables are normally distributed, must be used parametric test, if not, then must be used non-parametric test methods.

-  Normally distributed = t-test for comparison between independent samples (N =2) groups, ANOVA (N=3), paired t-test for comparison within a group between specific time points.

- Not normally distributed = Mann–Whitney U test between independent samples (N =2), Kruskal-Wallis (N=3), Wilcoxon-ranked Test for paired samples.

Section 4.7. analytical conditions and the respective subsections 4.7.1. and 4.7.2 should be renumbered correctly to 4.8 analytical conditions and the respective subsections 4.8.1 and 4.8.2.

Which were the criteria for the establishment of the GL-V9 concentrations to the in vitro assays and the doses in rat studies? These concentrations are expected to be therapeutic?

Author Response

Response to reviewer 3 comments:

The current manuscript describes the identification and quantification of a glucuronide metabolite of GL-V9 in vitro and in vivo and investigation of the major isoforms of 5 UDP-glucuronosyltransferase (UGT) involved. The manuscript seems to be well written and methodologically sound. However, some points should be clarified or ameliorated:

Point 1: Section 2.5: other pharmacokinetic parameters should be presented (e.g . Cmax, tmax, AUC0-inf, t1/el, etc). How was performed the oral delivery? Which was the reason to select the oral delivery as it was expected a low bioavailability?

Response 1: We highly appreciate the reviewer’s thoughtful suggestions. The other pharmacokinetic parameters of GL-V9, including Cmax, Tmax, AUC0-inf, T1/2, were reported in the previous work of our group which to be published. And the pharmacokinetic parameters of 5-O-glucuronide GL-V9 were listed in the manuscript.

The oral delivery to rats was performed using lavage needle. As for the selection of administration routes for GL-V9, we considered a variety of factors including the metabolism pathway for flavonoids, the properties of wogonin (its parent compound), and the future clinical disposition. For most of flavonoids, including wogonin, the first-pass metabolism after oral administration is the main cause accounted for their undesirable bioavailability and limits their clinical development [1–3]. What’s more, the aim of design and generation of GL-V9 is to modify the structure of wogonin and thus hurdle the problem of poor bioavailability, but the result was not desirable that GL-V9 also underwent a high level of metabolism by liver after oral delivery [4]. Therefore, we select the oral route to investigate its disposition and metabolism in vivo, hoping to find more useful information to support the further development of this kind of flavonoids.

Point 2: Section 4.3: the centrifugation speed should be in g (not in rpm).

Response 2: We appreciate the valuable comment from reviewer. We didn’t think about it before, but actually “g” is a more scientific unit to descript the centrifugation speed. Thus, we have changed “rpm” to “g” in the manuscript.

Point 3: Section 4.8: which was the statistical test used, one-way ANOVA or student-t test? The authors tested for normal distribution? When variables are normally distributed, must be used parametric test, if not, then must be used non-parametric test methods.

Response 3: We are very sorry for our negligence of this error and highly appreciate the reviewer’s observant suggestion. Only one-way ANOVA with Tukey test was used to carry out the statistical analysis in present study, and all the data were tested for normally distribution before the statistical test. This error has been revised in the manuscript and changes are highlighted using the “Track changes”.

Point 4: Section 4.7. analytical conditions and the respective subsections 4.7.1. and 4.7.2 should be renumbered correctly to 4.8 analytical conditions and the respective subsections 4.8.1 and 4.8.2.

Response 4: Thanks a lot for reviewer’s careful work on our manuscript. And the numbers have been corrected in the main text.

Point 5: Which were the criteria for the establishment of the GL-V9 concentrations to the in vitro assays and the doses in rat studies? These concentrations are expected to be therapeutic?

Response 5: The concentrations of GL-V9 used in in vitro experiments were selected based on the pharmacodynamic data published by Li et al. [5,6], which was equal to the Cmax in PK studies. Anti-tumor effects of GL-V9 were optimal in micromolar range as reported in afore-mentioned publications. Moreover, with in house data (not published) from our group, the selected oral dosing was able to maintain GL-V9 concentrations in major organs at therapeutic-relevant range.

References:

1.    Walle, T.; Otake, Y.; Brubaker, J.A.; Walle, U.K.; Halushka, P.V. Disposition and metabolism of the flavonoid chrysin in normal volunteers. British Journal of Clinical Pharmacology 2001, 51, 143–146.

2.    Walle, T. High absorption but very low bioavailability of oral resveratrol in humans. Drug Metabolism and Disposition 2004, 32, 1377–1382.

3.    Otake, Y. Glucuronidation versus Oxidation of the Flavonoid Galangin by Human Liver Microsomes and Hepatocytes. Drug Metabolism and Disposition 2002, 30, 576–581.

4.    Xing, H.; Ren, C.; Kong, Y.; Ning, C.; Kong, D.; Zhang, Y.; Zhao, D.; Li, N.; Wang, Z.; Chen, X.; et al. Mechanistic study of absorption and first‐pass metabolism of GL‐V9, a derivative of wogonin. Biopharmaceutics & Drug Disposition 2019.

5.    Li, L.; Lu, N.; Dai, Q.; Wei, L.; Zhao, Q.; Li, Z.; He, Q.; Dai, Y.; Guo, Q. GL-V9, a newly synthetic flavonoid derivative, induces mitochondrial-mediated apoptosis and G2/M cell cycle arrest in human hepatocellular carcinoma HepG2 cells. European Journal of Pharmacology 2011, 670, 13–21.

6.    Li, L.; Chen, P.; Ling, Y.; Song, X.; Lu, Z.; He, Q.; Li, Z.; Lu, N.; Guo, Q. Inhibitory effects of GL-V9 on the invasion of human breast carcinoma cells by downregulating the expression and activity of matrix metalloproteinase-2/9. European Journal of Pharmaceutical Sciences 2011, 43, 393–399.

Reviewer 4 Report

In this study, the authors identified glucuronidation of GL-V9 in vitro and in vivo. The authors found 5-O-glucuronide GL-V9 is the main form of GL-V9 glucuronidation. Based on this, the authors performed systematical analysis of glucuronidation of GL-V9 in 5 different species and via 12 different human UGTs. The authors provided detailed profiling of GV-V9 metabolites in different species and different tissues. The manuscript was written well and experiments were well designed and performed.
Minor revisions:
 1) The authors should use abbreviation properly, For example, indicating what are DLM, RLM, HLM before using them in main text or figures.
 2) Figure 5, missing A, B, C, D... labels for each panel.
 3) Figure 7, the authors should provide a zoom-in figure of plots for the early time points.
 4) Some legends are too short and not well written. The authors should re-write the legends for the following figures: Figure 2, 5, 6 and 7.

Author Response

Response to the reviewer 4 comments:

In this study, the authors identified glucuronidation of GL-V9 in vitro and in vivo. The

authors found 5-O-glucuronide GL-V9 is the main form of GL-V9 glucuronidation.

Based on this, the authors performed systematical analysis of glucuronidation of GLV9

in 5 different species and via 12 different human UGTs. The authors provided

detailed profiling of GV-V9 metabolites in different species and different tissues. The

manuscript was written well and experiments were well designed and performed.

Minor revisions:

Point 1: The authors should use abbreviation properly, For example, indicating what are DLM, RLM, HLM before using them in main text or figures.

Response 1: We highly appreciate the reviewer’s valuable suggestions pointing out the writing issues in our manuscript. The explanations for these abbreviations, including ‘DLM, RLM, HLM, MiceLM, and MLM’, have been added at the site where it first occurred in the revised manuscript.

Point 2: Figure 5, missing A, B, C, D... labels for each panel.

Response 2: Thank you very much for your careful work on reviewing our manuscript. We are sorry for our negligence of missing labels in Figure 5, and these labels for each panel have been added in the redrawn figure.

Point 3: Figure 7, the authors should provide a zoom-in figure of plots for the early time points.

Response 3: We highly appreciate the valuable suggestion from reviewer, which aims to improve the quality of the figure and provide a better presentation of results to readers. The Figure 7 has been redrawn with a zoom-in figure of plots before 4 h post-dose.

Point 4: Some legends are too short and not well written. The authors should re-write the legends for the following figures: Figure 2, 5, 6 and 7.

Response 4: We really appreciate the valuable suggestion from reviewer. The legends for all figures have been rephrased and extended to provide more details about experiments and results of the present study in the revised manuscript.

Reviewer 5 Report

In this study, the involvement of UGT enzymes regarding the biotransformation of the wogonin derivative GL-V9 was investigated, considering isoforms and species differences. The topic of the study is interesting, and the experimentation seems appropriate. The conclusions are supported by the results; however, some further discussion need to be inserted into the manuscript. Figures and corresponding labels (e.g., axes) need to be improved for the appropriate quality. The manuscript was written in a poor linguistic style and contains several typos, which obviously need to be corrected. The manuscript describes a valuable work; however, I can suggest its publication after a revision. My major comments are listed below.

Significant correction of the linguistic style is reasonable. Furthermore, Authors have to check typos.

It would be useful to prepare a figure (Fig. 1) which contains the chemical structure of 5-O-glucuronide of GL-V9. Please mark with different colors the structure of wogonin, its chemical modification (results in the final structure of GL-V9), and the glucuronic acid part.

Lines 20 and 24: It is unnecessary to discuss that the 5-O-glucuronide is the main glucuronide metabolite of GL-V9 because GL-V9 possesses only one hydroxyl group (in position 5)… (as it is also described in the Discussion section)

Lines 45, 49, etc: “Anticancer” word should be used only if the compound has proved in vivo action against cancer.

Lines 52-57: References are missing.

Lines 60-62: “As GL-V9 is the derivative of wogonin and shared the same backbone with it, it is conjectured that the metabolites of GL-V9 may also have the influence on the therapeutic effect of GL-V9.” – This statement is very speculative. Justify or clear.

Lines 71-73: Please specify the species regarding these statements.

Figs. 1-7: Figures and texts included (e.g., axes) are too small. Size and quality need to be significantly improved.

Figs. 4-7 and tables 1-2: Statistics is missing. Authors have to demonstrate the result of the statistical analyses in figures and tables.

Figs. 5-6: Give more detailed captions to these figures.

Fig. 7: It is not clear for me: Both the parent compound and the glucuronide were administered orally (both 50 mg/kg), or only the parent compound? If both compounds were administered, Authors should give a more detailed discussion of these results. It is very likely that the glucuronide metabolite will not be absorbed only after its microbial hydrolysis…

Lines 211-217: I would like to refer back to my third comment (regarding lines 20 and 24). Since the 5-hydroxyl group is relatively close to the 4-carbonyl group, it is reasonable to hypothesize the formation of hydrogen bond, which can stabilize the H. Furthermore, the bulky structure of glucuronic acid may also explain why the substitution in position 5 is not favorable if other hydroxyl groups (e.g., in position 7) are also available. Since GL-V9 possesses only one hydroxyl group located in position 5, obviously, only 5-O-glucuronide derivative of GL-V9 can be formed as its glucuronic acid metabolite. Therefore, I suggest to rephrase and slightly extend this section, using more references from the literature.

Lines 217-219: “we demonstrated that 5-O-glucuronide GL-V9 was the main phase II metabolite of GL-V9 with combining in vitro and in vivo results” – This statement is not established because only glucuronide metabolites were tested in this study. Hypothetically other (e.g., sulfate or methyl) conjugates may also appear in significant concentrations.

Lines 247-248: “The results demonstrated that the formation of metabolite was always consistent with the concentration of the parent compound.” – I am not sure what this sentence really means. Please rephrase.

Why the doses administered are different in 4.3 and 4.6 (100 and 50 mg/kg, respectively)?

Line 316: “reaction system and operations are as same as above” – Please specify the corresponding section.

Section 4.8: Which statistical analysis was applied (ANOVA or t-test) in which part of the study? Which post-hoc test was applied regarding ANOVA? Was the same level of significance (p0.05) applied in both ANOVA and t-test? 

Author Response

Response to the reviewer 5 comments:

In this study, the involvement of UGT enzymes regarding the biotransformation of the wogonin derivative GL-V9 was investigated, considering isoforms and species differences. The topic of the study is interesting, and the experimentation seems appropriate. The conclusions are supported by the results; however, some further discussion need to be inserted into the manuscript. Figures and corresponding labels (e.g., axes) need to be improved for the appropriate quality. The manuscript was written in a poor linguistic style and contains several typos, which obviously need to be corrected. The manuscript describes a valuable work; however, I can suggest its publication after a revision. My major comments are listed below.

Point 1: Significant correction of the linguistic style is reasonable. Furthermore, Authors have to check typos.

Response 1: We highly appreciate the suggestion from the reviewer which aims to improve the quality of this manuscript, and feel sorry for our language errors. We have carefully revised the manuscript, checked and corrected typos according to the reviewer’s comments on English writing.

Point 2: It would be useful to prepare a figure (Fig. 1) which contains the chemical structure of 5-O-glucuronide of GL-V9. Please mark with different colors the structure of wogonin, its chemical modification (results in the final structure of GL-V9), and the glucuronic acid part.

Response 2: We really appreciate the constructive suggestions on our manuscript. The chemical structures in Fig.1 and Fig.2 have been redrawn in different colors to highlight the chemical modification (in red) and the glucuronic acid part (in blue).

Point 3: Lines 20 and 24: It is unnecessary to discuss that the 5-O-glucuronide is the main glucuronide metabolite of GL-V9 because GL-V9 possesses only one hydroxyl group (in position 5)… (as it is also described in the Discussion section)

Response 3: We really appreciate the reviewer’s valuable comment. This section in the abstract has been rephrased and the changes are highlighted in red font in the revised manuscript.

Point 4: Lines 45, 49, etc: “Anticancer” word should be used only if the compound has proved in vivo action against cancer.

Response 4: We appreciate the reviewer’s valuable suggestion. We didn’t think about it before, but actually “anticancer” is not suitable here to describe the bioactivity of GL-V9. Thus, we changed the “anticancer” to “anti-tumor” in the manuscript.

Point 5: Lines 52-57: References are missing.

Response 5: We feel so sorry for our negligence and thanks for the reminder form the reviewer. The references related to these statements have been added in the revised manuscript.

Point 6: Lines 60-62: “As GL-V9 is the derivative of wogonin and shared the same backbone with it, it is conjectured that the metabolites of GL-V9 may also have the influence on the therapeutic effect of GL-V9.” – This statement is very speculative. Justify or clear.

Response 6: We highly appreciate the reviewer’s valuable comment. The GL-V9 is a kind of wogonin derivatives with chemical modification, and the same backbone endows them similar properties like bioactivities and metabolism pathway. Because the glucuronide metabolite of wogonin, wogonoside, showed significant anti-tumor activity in the previous studies [1-3], we speculated that the glucuronide metabolite of GL-V9 may also have the bioactivities and thus it’s meaningful to carry out this systemic metabolism study of GL-V9. We have rephrased this statement in the revised manuscript hoping to clarify the reason why we carried out a series of experiments to investigate the glucuronidation metabolism of GL-V9.

Point 7: Lines 71-73: Please specify the species regarding these statements.

Response 7: Thank you for your careful work. All these statements described the expression of UGTs in human liver. And we have specified the species in the revised manuscript.

Point 8: Figs. 1-7: Figures and texts included (e.g., axes) are too small. Size and quality need to be significantly improved.

Response 8: We highly appreciate reviewer’s instructive comment. All figures in the manuscript have been redrawn to improve the quality, and we hope that will help us to better clarify the study.

Point 9: Figs. 4-7 and tables 1-2: Statistics is missing. Authors have to demonstrate the result of the statistical analyses in figures and tables.

Response 9: We really appreciate reviewer’s valuable comment. We didn’t think about demonstrating the statistical results in figures and tables before, and we only discussed that in the main text. Now, the statistic differences were also shown as “*” in figures and tables in the revised manuscript.

Point 10: Figs. 5-6: Give more detailed captions to these figures.

Response 10: We highly appreciate reviewer’s instructive suggestion. The captions of all these figures in the manuscript have been rephrased and more details about experiments and results have been presented in the captions.

Point 11: Fig. 7: It is not clear for me: Both the parent compound and the glucuronide were administered orally (both 50 mg/kg), or only the parent compound? If both compounds were administered, Authors should give a more detailed discussion of these results. It is very likely that the glucuronide metabolite will not be absorbed only after its microbial hydrolysis…

Response 11: Only the parent compound, GL-V9, was administered orally to rats. The metabolite of GL-V9 has never been identified and investigated before, and it’s also the first time for the 5-O-glucuronide GL-V9 to be identified and generated through chemical synthesis. In fact, there are not sufficient evidences to prove that the 5-O-glucuronide GL-V9 is safe and non-toxic for rats or possesses any bioactivity, thus it is unreasonable to investigate it in vivo after administration of this compound to rats at the present stage of research.

Point 12: Lines 211-217: I would like to refer back to my third comment (regarding lines 20 and 24). Since the 5-hydroxyl group is relatively close to the 4-carbonyl group, it is reasonable to hypothesize the formation of hydrogen bond, which can stabilize the H. Furthermore, the bulky structure of glucuronic acid may also explain why the substitution in position 5 is not favorable if other hydroxyl groups (e.g., in position 7) are also available. Since GL-V9 possesses only one hydroxyl group located in position 5, obviously, only 5-O-glucuronide derivative of GL-V9 can be formed as its glucuronic acid metabolite. Therefore, I suggest to rephrase and slightly extend this section, using more references from the literature.

Response 12: We highly appreciate this instructive suggestion from the reviewer and agree with your point. In order to fully elaborate this section, we referred to more literature and found that the occurrence of glucuronidation at 5-OH group in wogonin was reported by Chenrui Li et al [10]. Actually, there are two hydroxyl groups presented in the structure of wogonin at 5-position and 7-position. However, the 5-O-glucuronide metabolite was also found in rats after oral administration, confirming the reaction activity of the 5-OH group. Combined with the fact that GL-V9 only possesses one hydroxyl group located at 5-position, we speculated that its glucuronide metabolite can only be the 5-O-glucuronide GL-V9, which was then confirmed by our experimental results. The statements in the revised manuscript about this section have also been rephrased and extended according to the reviewer’s comment.

Point 13: Lines 217-219: “we demonstrated that 5-O-glucuronide GL-V9 was the main phase II metabolite of GL-V9 with combining in vitro and in vivo results” – This statement is not established because only glucuronide metabolites were tested in this study. Hypothetically other (e.g., sulfate or methyl) conjugates may also appear in significant concentrations.

Response 13: Thanks a lot for your valuable advice, and it’s our negligence that did not mention and discuss the results of our work on finding other kinds of conjugates in the manuscript. In fact, we didn’t find any other phase II metabolite except 5-O-glucuronide GL-V9 both in vivo and in vitro. We also calculated the metabolism fraction of 5-O-glucuronide GL-V9 according to the quantification study in rats which was described in section ‘2.5 Quantification of GL-V9 and 5-O-glucuronide GL-V9 in vivo’. The result showed that the metabolism fraction of it was nearly 88%, indicating that most of GL-V9 were converted into 5-O-glucuronide conjugates after oral delivery. Therefore, we demonstrated that the 5-O-glucuronide GL-V9 is the main phase II metabolite of GL-V9.

Point 14: Lines 247-248: “The results demonstrated that the formation of metabolite was always consistent with the concentration of the parent compound.” – I am not sure what this sentence really means. Please rephrase.

Response 14: We are sorry for nor descripting it clearly, and this sentence has already been rephrased in the revised manuscript as “As shown in Figure 7, formation of 5-O-glucuronide GL-V9 generally corresponded with the elimination of parent GL-V9 at each measured time point. These results indicated that GL-V9 underwent a rapid and extensive biotransformation to 5-O-glucuronide upon absorption”.

Point 15: Why the doses administered are different in 4.3 and 4.6 (100 and 50 mg/kg, respectively)?

Response 15: We really appreciate for your careful review on our manuscript. In the section 4.3, which aims to find out and identify the metabolites of GL-V9 in vivo, we chose a higher administration dose (100 mg/kg) to ensure the detectable amount of metabolites in the rats’ plasma and excreta. While for the quantification study in section 4.6, the dose (50 mg/kg) is the same as that in PK study of GL-V9 carried out by our group previously (data to be published), which was based on the pharmacodynamical researches reported by Liwen Li et al [5,6].

Point 16: Line 316: “reaction system and operations are as same as above” – Please specify the corresponding section.

Response 16: Thank you a lot to point out the inappropriate description in our manuscript. The corresponding section ‘4.3’ has been added to this sentence in the revised manuscript.

Point 17: Section 4.8: Which statistical analysis was applied (ANOVA or t-test) in which part of the study? Which post-hoc test was applied regarding ANOVA? Was the same level of (p<significance 0.05) applied in both ANOVA and t-test?.

Response 17: We feel so sorry for our negligence of this error and highly appreciate the reviewer’s instructive comment. Only one-way ANOVA with Tukey test was used to carry out the statistical analysis in the present study, and the significant difference was defined as p0.05. This error has been revised in the manuscript and changes are highlighted using the “Track changes”.

References:

1.   Han, C.; Xing, G.; Zhang, M.; Zhong, M.; Han, Z.; He, C.; Liu, X. Wogonoside inhibits cell growth and induces mitochondrial-mediated autophagy-related apoptosis in human colon cancer cells through the PI3K/AKT/mTOR/p70S6K signaling pathway. Oncology Letters 2018.

2.   Li, Y.; Tu, M.; Cheng, C.; Tian, J.; Zhang, F.; Deng, Z.; Li, X.; Li, Z.; Liu, Y.; Lei, G. Wogonoside induces apoptosis in Bel-7402, a hepatocellular carcinoma cell line, by regulating Bax/Bcl-2. Oncology Letters 2015, 10, 1831–1835.

3.   Zhang, L.; Wang, H.; Cong, Z.; Xu, J.; Zhu, J.; Ji, X.; Ding, K. Wogonoside induces autophagy-related apoptosis in human glioblastoma cells. Oncology Reports 2014, 32, 1179–1187.

4. Li, C.; Zhang, L.; Lin, G.; Zuo, Z. Identification and quantification of baicalein, wogonin, oroxylin A and their major glucuronide conjugated metabolites in rat plasma after oral administration of Radix scutellariae product. Journal of Pharmaceutical and Biomedical Analysis 2011, 54, 750–758.

5.   Li, L.; Lu, N.; Dai, Q.; Wei, L.; Zhao, Q.; Li, Z.; He, Q.; Dai, Y.; Guo, Q. GL-V9, a newly synthetic flavonoid derivative, induces mitochondrial-mediated apoptosis and G2/M cell cycle arrest in human hepatocellular carcinoma HepG2 cells. European Journal of Pharmacology 2011, 670, 13–21.

6.   Li, L.; Chen, P.; Ling, Y.; Song, X.; Lu, Z.; He, Q.; Li, Z.; Lu, N.; Guo, Q. Inhibitory effects of GL-V9 on the invasion of human breast carcinoma cells by downregulating the expression and activity of matrix metalloproteinase-2/9. European Journal of Pharmaceutical Sciences 2011, 43, 393–399.

Round 2

Reviewer 2 Report

OK

Reviewer 5 Report

The authors significantly improved the manuscript and answered my questions. I have no further critical comments.